# Inhibitors, PROTACs and Molecular Glues as Diverse Therapeutic Modalities to Target Cyclin-Dependent Kinase

**DOI:** 10.3390/cancers13215506

**Published:** 2021-11-02

**Authors:** Sandeep Rana, Jayapal Reddy Mallareddy, Sarbjit Singh, Lidia Boghean, Amarnath Natarajan

**Affiliations:** 1Division of Preclinical Innovation, National Center for Advancing Translational Sciences, National Institutes of Health, Rockville, MD 20850, USA; sandeep.rana@nih.gov; 2Eppley Institute for Research in Cancer and Allied Diseases, University of Nebraska Medical Center, Omaha, NE 68198, USA; j.mallareddy@unmc.edu (J.R.M.); sarbjit.singh@unmc.edu (S.S.); lidia.boghean@unmc.edu (L.B.); 3Pharmaceutical Sciences and University of Nebraska Medical Center, Omaha, NE 68198, USA; 4Genetics Cell Biology and Anatomy, University of Nebraska Medical Center, Omaha, NE 68198, USA; 5Fred and Pamela Buffett Cancer Center, University of Nebraska Medical Center, Omaha, NE 68198, USA

**Keywords:** kinase, CDK, cancer, PROTAC, molecular glue

## Abstract

**Simple Summary:**

Cyclin-dependent kinases (CDKs) are rich and viable therapeutic targets for various cancers. The emergence of event-driven pharmacology as an alternative to occupancy-driven pharmacology has begun to address the challenges associated with selectively targeting CDKs. In this review article, we summarize the CDK inhibitors that are currently in clinical trials. In addition, we provide an overview of PROTAC- and molecular glue-based strategies to modulate CDK function.

**Abstract:**

The cyclin-dependent kinase (CDK) family of proteins play prominent roles in transcription, mRNA processing, and cell cycle regulation, making them attractive cancer targets. Palbociclib was the first FDA-approved CDK inhibitor that non-selectively targets the ATP binding sites of CDK4 and CDK6. In this review, we will briefly inventory CDK inhibitors that are either part of over 30 active clinical trials or recruiting patients. The lack of selectivity among CDKs and dose-limiting toxicities are major challenges associated with the development of CDK inhibitors. Proteolysis Targeting Chimeras (PROTACs) and Molecular Glues have emerged as alternative therapeutic modalities to target proteins. PROTACs and Molecular glues utilize the cellular protein degradation machinery to destroy the target protein. PROTACs are heterobifunctional molecules that form a ternary complex with the target protein and E3-ligase by making two distinct small molecule–protein interactions. On the other hand, Molecular glues function by converting the target protein into a “*neo-substrate*” for an E3 ligase. Unlike small molecule inhibitors, preclinical studies with CDK targeted PROTACs have exhibited improved CDK selectivity. Moreover, the efficacy of PROTACs and molecular glues are not tied to the dose of these molecular entities but to the formation of the ternary complex. Here, we provide an overview of PROTACs and molecular glues that modulate CDK function as emerging therapeutic modalities.

## 1. Introduction

Protein kinases are a family of essential enzymes that make up ~2.6% of all human genes (based on estimated 20,000 human genes). They are clustered in different subgroups such as serine-threonine (S/T) kinases, tyrosine kinases, and tyrosine-like kinases, based on their sequence similarity and biochemical function [1]. Mechanistically, a kinase forms a ternary complex with ATP and its substrate to catalyze the transfer of the γ-phosphate group from an ATP to its substrate proteins. The resulting phosphorylation of the substrate serves as the on/off switch for downstream signal transduction pathways. Unlike the classical kinases, histidine kinases are two-component signal transduction systems, i.e., the transfer of the γ-phosphate group from an ATP to the histidine residue of the kinase, followed by the transfer of phosphate to the aspartate residue of the downstream substrate [2,3].

Changes to the phosphoproteome regulate several aspects of cellular functions, such as growth, migration, and survival, etc. Therefore, dysregulation of the kinase function has been associated with the pathogenesis of several human diseases, including cancer and neurodegenerative diseases [4,5,6]. Therapeutically, a small, well-defined ATP binding pocket makes kinases an attractive target for drug discovery. Twenty years ago, Imatinib (Gleevec), a BCR-Abl kinase inhibitor, was the first Food and Drug Administration (FDA)-approved kinase inhibitor to treat Philadelphia chromosome-positive chronic myelogenous leukemia. As of 30 March 2021, 65 kinase inhibitors have been approved by the FDA, and 85% of them are prescribed to treat cancer [7].

A protein kinase core is composed of two lobes; a smaller N-terminal lobe that mainly consists of five β-sheets and a C_α_-helix, and a larger C-terminal lobe that primarily consists of six α-helices. These two lobes are connected by a flexible hinge region that interacts with the ATP [8]. On the other side of the hinge region is a flexible activation loop that begins with the conserved DFG motif and extends up to the APE motif. In the active conformation, the Asp (DFG in) residue orients inwards towards the ATP binding site and binds to the magnesium ion that interacts with the ATP phosphate. In the inactive conformation, the Asp (DFG out) residue points away from the ATP binding site. Kinase inhibitors that bind to the active kinase conformation are called type I inhibitors while those that bind to the inactive conformation are called type II inhibitors. Representative examples of type I and type II inhibitors are Gefitinib, Baricitinib, Ribociclib and Imatinib, Ponatinib, Sorafenib, respectively. Type III kinase inhibitors, such as Cobimetinib, Trametinib, and Binimetinib bind to an allosteric site of the kinase. Covalent kinase inhibitors bind irreversibly to the kinase by forming a carbon–sulfur bond through the sulfhydryl group of the surface-exposed cysteine residue proximal to the ATP binding pocket site [4].

To date, ~89% of approved kinase inhibitors are reversible and follow an occupancy-driven pharmacology model wherein the efficacy of the inhibitor correlates with its off constant [9]. Since it is impossible to accurately model the amount of drug required to maximally inhibit kinase function in vivo, efficacy studies are conducted at concentrations as close to the maximum tolerated dose as possible. Irreversible or covalent inhibitors that follow a two-step binding model overcome the above issue. First, the reversible component of the covalent inhibitor binds to the kinase (K_i_), followed by the attack of the proximal nucleophile with an appropriate trajectory to the electrophilic component of the covalent inhibitor (K_inact_) [10,11].

Cyclin-dependent kinases (CDKs) are members of a S/T kinase subfamily that comprises 21 CDK enzymes. CDK1, 2, 3, 4, and 6 play a key role in cell cycle regulation, whereas CDK7, 8, 9, and 11 are involved in the regulation of transcription. The biological functions of CDK10, 11, 14–18, and 20 are yet to be fully elucidated [12,13]. Cyclins bind to CDKs to regulate their activity, except CDK5, which binds to and is regulated by p35/p39 and their cleaved products p25/p29 [14]. Multiple cyclins bind to the same CDK to regulate different phases of the cell cycle. For instance, the CDK2-cyclin E complex regulates cell cycle reentry, G1 progression, and S phase entry, whereas CDK2-cyclin A regulates the later part, S and G2/M progression, of the cell cycle [15]. CDKs bind to different cyclins, resulting in the phosphorylation of an array of substrates [16]. For example, CDK9 can complex with either cyclin T1, T2a, T2b, or K to form the positive transcription elongation factor b (P-TEFb), which phosphorylates the C-terminal domain of the large subunit of RNA polymerase II (RNA pol II) and facilitates mRNA synthesis [13,14]. However, the CDK9–cyclin K complex has additional functions in genome maintenance, [15] and induces the phosphorylation of Ser33 within the transactivation domain (TAD) of p53 [16].

CDKs play important roles in a variety of functions including cell proliferation, transcriptional activity, and neuronal activity, and dysregulation of these proteins is often observed in cancer and neurological diseases [17]. Over the past two decades, several CDK inhibitors (**1–13**) entered clinical trials to treat cancer (Figure 1). In 2015, Palbociclib was the first CDK 4/6 inhibitor (**6**) to be approved by the FDA, [18] and shortly after, the CDK4/6 inhibitors ribociclib (**7**) [19] and abemaciclib (**8**) [20] also received FDA approval for the treatment of breast cancer. The above successes reinvigorated CDK inhibitor development, leading to several CDK inhibitors currently being evaluated alone or in combination with other drugs against several cancers (Table 1) [21].

Novel approaches to target dysregulated proteins are routinely explored. One such approach that has gained considerable attention is proteolysis targeting chimeras (PROTACs). PROTACs artificially induce the degradation of a targeted protein by hijacking the cellular quality-control machinery [22]. PROTACs are heterobifunctional molecules with two different ligands connected by a linker. One ligand attaches to the protein of interest (POI), whereas the other ligand binds to an E3 ligase, resulting in a ternary complex between POI, the PROTAC, and the E3 ligase. This enables proximity-driven polyubiquitination of the POI by the E3 ligase and the subsequent non-natural degradation of the target protein (Figure 2). A traditional small molecule inhibitor (SMI) primarily blocks the catalytic activity of the protein to inhibit the enzymatic function of the target protein via an occupancy-driven pharmacology. On the other hand, a PROTAC operates by degrading the target protein via an event-driven pharmacology model. Early studies showed superior effects via the PROTAC approach, likely due to the elimination of both catalytic and non-catalytic/scaffolding function of the target protein [23,24,25,26,27,28,29,30].

The developmental arc of PROTACs went from a proof-of-concept study to clinical trials in ~20 years (Figure 3). A typical PROTAC consists of three parts: a POI binding ligand, an E3 ubiquitin-binding ligand, and a linker connecting two ligands. Despite the presence of >600 putative E3 ligases in the human proteome, only a handful of E3 ligase ligands (CRBN [31,32], VHL [33,34,35], MDM [36,37,38], IAP [39], RNF114 [40,41], DCAF15 [42], DCAF 16 [43], and FEM1B [44]) are readily available for the design of a PROTAC (Figure 4). The PROTAC approach has been successfully applied to degrade kinases, G protein-coupled receptors, nuclear receptors, membrane proteins, transcription factors, and neurodegenerative proteins aggregates [23,26,27,28,29,30,36,45,46,47,48,49,50,51,52,53,54,55,56,57,58,59,60,61]. Recent studies showed good potency and selectivity of PROTACs in animal models due to POI degradation [27,28,29,30]. Currently, two PROTAC candidates, ARV-471 (**14**, NCT04072952) and ARV-110 (**15**, NCT03888612), targeting the estrogen receptor and androgen receptor, respectively, are in Phase II clinical trials. Two additional PROTACs, viz. KT-474 (undisclosed structure) (NCT04772885) that degrades IRAK4 for the treatment of interleukin-1 receptor (IL-1R)/toll-like receptor (TLR)-driven immune-inflammatory diseases, and BTK degrader NX-2127 (NCT04830137) for the treatment of relapsed and refractory B-cell malignancies, are being evaluated in Phase I studies.

A typical flowchart for the successful development of a PROTAC is summarized in Figure 5. The PROTAC strategy has been effective in converting promiscuous binders to selective degraders [62,63,64]. Due to the catalytic nature of PROTAC technology, a sub-stoichiometric amount of PROTAC is adequate to induce robust POI degradation, thereby mitigating the off-target inhibitory effects observed with promiscuous SMI. Since PROTACs degrade the POI, unlike SMI’s, PROTAC treatment does not result in feedback induction of the POI. Future studies will focus on developing disease-specific PROTACs either by exploring novel disease-specific POI such as oncogenic KRAS^G12C^, or by examining tissue-specific E3 ligases [65,66]. Alternative strategies to achieve tissue specificity could include the development of antibody PROTAC conjugates [67].

## 2. CDK Degradation Using PROTAC

In the following sections, we summarize PROTACs developed against various CDKs and, where available, compare their effects with the corresponding inhibitors. In general, the identification of a successful CDK PROTAC requires a set of generic experiments that are common such as: (i) binding studies with the newly synthesized PROTACs via in vitro kinase assays; (ii) dose–response studies to determine their cellular DC_50_; (iii) time-course studies to demonstrate the robust degradation of a CDK kinase at different time points; (iv) competition studies with an inhibitor, an E3 ligand, and a proteasome inhibitor to show the formation of a ternary complex and subsequent ubiquitin-mediated degradation of the CDK; and (v) whole cell proteomics studies to show selective degradation of the target CDK. In the subsequent sections, we provide the structures of selective CDK PROTACs along with an overview of experiments conducted to establish their MOA. Moreover, we highlight the CDK inhibitor component of a PROTAC with a yellow box and an E3 ligand component with a green box.

### 2.1. CDK2

Cyclin-dependent kinase 2 (CDK2) plays a crucial role in cell cycle regulation and other biological processes such as DNA damage, intracellular transport, and signal transduction [68]. CDK2 binds with cyclin E to phosphorylate numerous proteins required for the cell cycle progression, DNA replication, and centrosome duplication. During the S phase, cyclin E is degraded, and CDK2 then forms complexes with cyclin A [69,70,71]. Although CDK2 mutations are rarely observed in human cancers, the CDK2/cyclin E and CDK2/cyclin A pathways are often altered. For instance, in ovarian and breast cancers, cyclin E1 gene amplification results in increased activity of CDK2/cyclin E complexes [72,73]. Similarly, cyclin A is often overexpressed in colorectal [74], breast [75], and high-grade serous ovarian cancers (HGSOC) [76]. Recent studies also suggest that inhibiting CDK2 could be a promising therapeutic strategy to treat acute myeloid leukemia (AML) [77,78]. Despite extensive efforts put forth by both pharma and academic labs, the discovery of a highly selective CDK2 inhibitor remains elusive, primarily due to the high homology of the ATP binding sites among the CDKs [68,79].

The PROTAC strategy relies on the overall three-dimensional architecture of CDK and involves structural complementarity with the E3 ligase complex. This approach has been explored to develop a selective CDK2 degrader. Three different labs reported the development of CDK2 PROTACs using pomalidomide (**16**) and three different CDK targeting ligands (**27**, **29** and **31**, Figure 6). OVCAR8 cells treated with PROTAC **27** (TMX-2172) selectively degraded CDK2 and CDK5 over other CDKs [80]. The study also reported a control compound **28** (TMX-bump) in which the glutarimide nitrogen was methylated, thus blocking its ability to bind CRBN (E3 ligase). Ribociclib (**7**) does not inhibit CDK2; however, replacing the 1-(pyridyl-2-yl)piperazine with a methylbenzoate in ribociclib yielded an inhibitor that was selective of CDK2 (>50-fold) and bound CDK2 with improved affinity. PROTAC **29** derived from the modified ribociclib analog degraded CDK2 in addition to CDK4 and CDK6. PROTAC **29**, degraded CDK2/4/6 in melanoma cells (B16F10 and A375), and inhibited retinoblastoma (Rb) protein phosphorylation. Moreover, PROTAC **29** inhibited colony formation and induced apoptosis in A375 and B16F10 cells [81]. A prodrug **30** of PROTAC **29**, was synthesized by attaching a labile group on thalidomide which resulted in improvement of oral bioavailability from 1 to 68%. In a B16F10 xenograft model, oral administration (200 mg/kg) of **30** showed significant reduction in tumor growth. The third reported CDK2 PROTAC (**31**, Figure 6) was built using a CDK2 inhibitor with a diaminotriazole (J2) [82]. PROTAC **31** induced robust degradation of CDK2 in NB4 cells and Ramos cells at nanomolar concentrations. Accumulation of mutations within hematopoietic stem cells (HSCs) leads to AML [83]. In a cellular differentiation study, PROTAC **31** treatment induced the expression of CD11b, a marker for cell differentiation. Morphological changes indicated that PROTAC **31** could induce significant differentiation of myeloid cells [82]. Furthermore, PROTAC **31** induced granulocytic differentiation of HSCs, suggesting that PROTAC **31** is a promising lead for AML differentiation therapy.

### 2.2. CDK4

CDK4 and its close homolog CDK6 are a family of serine-threonine kinases that interact with D-type cyclins (D1, D2, and D3) to phosphorylate Rb to regulate the G1/S transition of the cell cycle [84]. Phosphorylation of Rb results in the release of transcription factors E2F to activate transcription of numerous genes that are responsible for cell cycle progression through the S phase as well as those involved in DNA replication and chromosome segregation [1]. Activation of the CDK4-Rb pathway is often observed in human cancers; for example, CDK4 amplification is observed in ~50% of glioblastomas [85]. Although CDK4 and CDK6 are highly homologous, they elicit different biological functions in cells. For instance, knockdown of CDK4 in the mouse model of NSCLC results in the CDK6-induced expression of mutant K-Ras [86] and CDK4/6 inhibitor treatment in breast cancer cells results in the amplification of CDK6 through a feedback mechanism [87]. These findings indicate the need to develop CDK4- and CDK6-selective inhibitors. The high homology (>90%) in the ATP binding pocket of CDK4 and CDK6 makes the development of CDK4- and CDK6-selective inhibitors a challenging task.

The synthesis and screening of a focused set of PROTACs derived from the CDK4/6 inhibitors palbociclib, ribociclib and abemaciclib identified selective and non-selective CDK4 and CDK6 degraders. The ribociclib-pomalidomide PROTAC **32** (Figure 6) with a 7-atom linker selectively degraded CDK4 in a CRBN-dependent manner in Jurkat cells [88]. Another study that evaluated a palbociclib and a ribociclib PROTAC linked through a triazole showed that the palbociclib-based PROTAC pal-pom (**33**) degraded CDK4 with three-fold selectivity over CDK6 in MDA-MB-231 cells [60]. Despite available CDK4/6 inhibitors and E3-ligase binding ligands, the development of fully characterized CDK4-selective degraders are yet to be achieved.

### 2.3. CDK6

The Gray lab, our lab and Rao lab independently discovered CDK6-selective PROTACs (**34**, **36** and **37**, Figure 7) by conjugating the linker through the solvent exposed piperazine on Palbociclib and functionalizing the 4-position of phthalimide. The three PROTACs had different linker lengths and linker compositions [46]. PROTAC **34** exhibited potent and selective CDK6 degradation, while the designed *N*-methylated thalidomide containing compound BSJ-bump (**35**) did not degrade CDK4 or CDK6. Both PROTAC **34** and parent inhibitor **6** induced a similar transcriptional response in AML cells, suggesting that the transcriptional role of CDK6 in AML is mainly limited to its kinase-dependent function. PROTAC **36** exhibited similar in vitro potency for CDK4 and CDK6. In MiaPaCa2 cells, PROTAC **36** with a linker length of 17 atoms selectively degraded CDK6 while sparing its close homolog CDK4. We also observed Hook effects [89] at 5 and 10 μM concentrations in both HPNE and MiaPaCa2 cell lines [90,91]. PROTAC **37** (CP-10) showed robust CDK6 degradation in human glioblastoma U251 cells [92]. Since CDK6 overexpression could result in resistance to CDK6 inhibitors, additional copies of CDK6 were introduced in A-673, an Ewing’s sarcoma cell line. In the above cell line, treatment with degrader **37** reduced CDK6 levels, suggesting that CDK6 PROTACs could be used to reverse CDK6 overexpression-induced drug resistance [92].

Multiple studies explored selectivity for CDK4 or CDK6 degradation by building focused sets of analogs with palbociclib, ribociclib and abemaciclib as the POI ligand, varying linker compositions, and E3-ligase ligands [60,93]. For the most part, PROTACs with shorter hydrophilic linker lengths and ribociclib as the POI ligand favored CDK4 degradation while those with longer hydrophobic linkers and palbociclib as the POI ligand resulted in selectivity for CDK6.

Steinebach and co-workers also explored the effect of linkers’ lengths and the type of E3 ligand on the activity and selectivity of PROTACs derived from CDK4/6 dual inhibitors [94]. Several PROTACs with varied linker lengths and E3 ligands i.e., CRL4^CRBN^, von Hippel Lindau (CRL2^VHL^), a cellular inhibitor of apoptosis protein 1 (cIAP1) and mouse double minute 2 homolog (MDM2) were synthesized [94]. The pomalidomide-based PROTAC **38** (Figure 7) reduced CDK4 and CDK6 levels in multiple myeloma (MM) cell lines at 100 nM concentration, but also degraded the Ikaros (IKZF1) and Aiolos (IKZF3) transcription factors. Interestingly, the effect of the linker length was not significant but the lipophilicity of PROTACs was crucial in determining its selectivity. In general, PROTACs with log D < 4, showed superior CDK6 degradation. Additionally, the PROTACs derived from different VHL ligands also showed pronounced degradation of CDK4 and CDK6 with some selectivity for CDK6. PROTACs synthesized using VHL ligands linked through the phenyl ring showed excellent selectivity toward CDK6 compared to those derived from the terminal amine group (**39** vs. **40, Figure 7**). Both PROTACs **39** and **40** were effective in degrading CDK6 in multiple cell lines. The IAP-based PROTAC **41** (Analog **35**, Figure 7) non-selectively degraded CDK4 and CDK6. However, the MDM2-based PROTAC was unable to degrade CDK4 and CDK6, probably due to their poor cellular permeability.

### 2.4. CDK8

CDK8, a serine/threonine protein kinase, is overexpressed in colorectal, breast, and hematological malignancies, and in colon cancer, high CDK8 levels are associated with poor clinical outcomes [95]. CDK8 knockdown by short hairpin RNA (shRNA) suppressed proliferation in both in vitro and in vivo models [96]. CDK8 plays a key role in oncogenic signaling pathways, including the Wnt/β-catenin pathway [96], the p53 pathway [97], the NOTCH signaling pathway [98], and the TGF-β signaling pathway [99]. CDK8 regulates transcription through both its kinase activity and its scaffolding function [100], therefore CDK8 PROTACs are an attractive therapeutic option [101].

CDK8 has a high sequence similarity (>90%) with CDK19 [102]. Cortistatin A is a potent ATP competitive CDK8/19 selective inhibitor [103]. The complex total synthesis of Cortistatin (16–30 steps), along with the poor overall yield (0.012 and 2%) [104,105,106], limited the development of Cortistatin A-based PROTACs. Based on the X-ray crystal structure and mutagenesis studies [107], Hatcher et al. designed an analog by replacing the complex core in Cortistatin A with a simple steroid scaffold such as dehydroepiandrosterone (DHEA). Further optimization of this core resulted in the development of a potent CDK8 inhibitor with IC_50_ of 17 nM. A PEG linker that conjugated CDK8 inhibitor to pomalidomide yielded PROTAC JH-XI-10-02 (**42**, Figure 7) that induced partial degradation of CDK8 in Jurkat cells at 1 µM [103]. Studies with a negative control PROTAC **43** (Analog **31**, Figure 7), that lacked the ability to bind CRBN, validated **42** as a CDK8-selective PROTAC.

### 2.5. CDK9

CDK9, a member of the positive transcription elongation factor b (P-TEFb) complex, associates with cyclin T and cyclin K to regulate transcriptional elongation by interacting with RNA polymerase II (RNAP II) [95]. Inhibition of CDK9 leads to reduced phosphorylation of RPB1, which results in a reduction in Mcl-1 levels. Mcl-1 is an anti-apoptotic protein that plays a crucial role in cancer cell survival [108,109,110]. Inhibition of CDK9 kinase activity resulted in the induction of apoptosis in several cancer cell lines [111,112,113,114,115]. These indicate that CDK9 is a promising cancer target, especially for cancers that are driven by transcriptional dysregulation [111,116].

We reported a PROTAC derived from a CDK2/5 aminopyrazole-based inhibitor that selectively degraded CDK9 without affecting the levels of CDK2, CDK5, AKT, FAK, and IKKβ [62,117]. In cellular studies, compound 3 reduced RPB1 phosphorylation, a direct substrate for CDK9, and reduced Mcl-1 levels. This was the first report that showed that the differential surface-exposed lysine residues among the CDKs could be exploited to convert a non-selective CDK inhibitor into a selective CDK9 degrader. In a follow-up study, we optimized linker length and linker composition to identify PROTAC 2 (**44**, Figure 8), which selectively degraded CDK9 with a DC_50_ value of 158 ± 6 nM [118]. Despite the remarkable selectivity for CDK9 degradation, as with compound 3, PROTAC **44** non-selectively inhibited CDK2, 5 and 9 in cell-free studies. Unlike the CDK9 inhibitor flavopiridol that reduced cyclin K levels, PROTAC **44** induced complete CDK9 degradation at 1 μM while sparing cyclin K. Recent studies demonstrated that concurrent inactivation of Mcl-1 and Bcl-xL resulted in robust induction of apoptosis [14,62,118,119,120] and CDK9 regulates the levels of pro-survival protein Mcl-1 [121,122]. Synergism studies showed PROTAC **44** sensitizes pancreatic cancer cells to Bcl2 inhibitors. PROTACs generated using two other aminopyrazole-based pan CDK inhibitors AT-7519, and FN-1501 resulted in CDK2/9 selective degraders. The FN-1501-based PROTAC induced G2/M arrest and inhibited the growth of PC3 cells with nanomolar potency [123].

Olson et al. developed a degrader, THAL-SNS-032 (**45, Figure 8**) by conjugating an N-acyl-2-aminothiazole CDK7/9 inhibitor **12 [124]** to pomalidomide via a PEG linker. PROTAC **45** induced potent and selective degradation of CDK9 in a CRBN-dependent manner [111]. In a cell-free assay, PROTAC **45** non-selectivity inhibited CDK1/2/7/9. PROTAC **45** selectively induced CDK9 degradation and did not alter the levels of CDKs inhibited by **12** even at 5μM. At higher concentrations, PROTAC **45** exhibited a hook effect as well [125]. Consistent with CDK9′s role in promoting transcriptional elongation, PROTAC **45** inhibited phosphorylation of Ser2 C-terminal domain (CTD) of RNA Polymerase II in dose- and time-dependent studies, without effecting Ser5 and Ser7 phosphorylation [111].

Multiple independent PROTAC development studies with different potent and selective CDK9 ligands yielded mixed results. Replacing the N-acyl-2-aminothiazole kinase binding ligand in **45** with NVP-2, a highly selective aminopyrimidine-derived CDK9 inhibitor with sub-nM potency for CDK9, resulted in weaker CDK9 degradation [111,126]. Wogonin is a selective CDK9 inhibitor with an IC_50_ of 0.19 µM [127]. A docking guided PROTAC library design identified 11c that showed sub-micromolar biochemical inhibition of CDK9/CycT1 [128], but degraded CDK9 with a DC_50_ ~ 10 μM and reduced Mcl-1 levels in a dose-dependent study [129,130]. Conjugating the CDK9-selective inhibitor BAY-1143572 through the solvent-exposed group with pomalidomide led to the identification of PROTAC B03 (**46, Figure 8**) [131]. In cell-free studies, PROTAC **46** inhibited CDK9 with IC_50_ ~ 8 nM, which was comparable to the parent inhibitor (IC_50_ ~ 13 nM). PROTAC **46** reduced the levels of CDK9 (DC_50_ ~ 7 nM) and Mcl-1 levels in a time-dependent study. Moreover, PROTAC **46** exhibited improved anti-proliferative activity when compared to its parent inhibitor [131]. Together, these results suggest that the efficacy of CDK9 degradation by various PROTACs does not correlate with the potency and selectivity for CDK9 inhibition by the parent inhibitor.

### 2.6. CDK12

CDK12 phosphorylates RNA polymerase II to regulate transcription elongation [132]. In addition, CDK12 has also been reported to play a critical role in RNA splicing, DNA damage response (DDR), and the maintenance of genomic stability [133,134]. Loss of *CDK12* results in reduced transcription of *BRCA1* and increases sensitivity to PARP inhibitors and platinum chemotherapy. Thus, CDK12 inhibitors could be used in combination with DNA-damaging agents in HR-deficient cancers [12]. CDK12 is overexpressed in a variety of cancers, including breast cancer, ovarian cancer, prostate cancer, and gastric cancer, making it a viable cancer target [132]. The development of CDK12 inhibitors is particularly challenging due to high sequence similarity with its close homolog CDK13. The Gray lab developed a CDK12-selective degrader, BSJ-4-116 (**47**, Figure 8), by attaching 3-aminopiperidine moiety, which was important for the selectivity for CDK12 over its close homolog CDK13, to a promiscuous multi kinase degrader TL12-186 [64,135]. In cellular assays, PROTAC **47** selectively degraded CDK12 in a dose- and a time-dependent study, while sparing CDK13. A robust ternary complex was observed between CDK12 and CRBN in the presence of **47**, which was abolished with inactive PROTAC BSJ-4-116-NC (**48**). On the other hand, no such complex was induced between CDK13 and CRBN. The control **48** was ten-fold less potent than **47** in inhibiting cell growth, indicating CDK12 degradation as the primary mode of action of **47**. Since CDK12 is implicated in the regulation of DDR genes, degrader **47** treatment resulted in the downregulation of transcription genes involved in DDR pathways. In Jurkat and MOLT4 cells, PROTAC **47** and PARP inhibitors showed strong synergism. Interestingly, chronic exposure of MOLT-4 and Jurkat cells to degrader **47** resulted in the development of CDK12 degradation resistance via mutations of Ile733Val and Gly739Ser on the CDK12 G-loop, which resulted in reduced degrader **47** binding.

## 3. Perturbing CDK Function by a Molecular Glue

In contrast to heterobifunctional PROTAC degraders (Figure 2), molecular glues are low molecular weight inducers (novel de novo contacts) or stabilizers of protein–protein interactions [136,137]. Upon binding to a protein, the small molecule induces a conformational change and makes the small molecule–protein complex a “*neo-substrate*” for E3 ligase (Figure 9). Following the formation of a ternary complex, the “*neo-substrate*” is ubiquitinated, which leads to UPS-mediated protein degradation. Although these glues were identified serendipitously, this is an emerging and exciting therapeutic strategy for the inactivation of intractable targets.

Thalidomide was prescribed to treat morning sickness in pregnant women in the late 1950s and was withdrawn in the early 1960s because of catastrophic teratogenicity. Studies conducted to understand the associated biology resulted in the characterization of Thalidomide as a molecular glue [138]. The mechanism of action of Thalidomide and its analogs, termed immune-modulatory imine drugs (IMiDs), involves the recruitment of the Ikaros zinc family transcription factor (IKZF1 and IKZF3), CK1α, GSPT1, and others to cullin-RING E3 ligase CUL4^CRBN^, resulting in their proteasomal degradation [31,32,139,140,141,142,143]. IMiDs, thalidomide, lenalidomide, and pomalidomide are first line therapeutics for multiple myeloma [144,145]. This discovery of IMiDs binding to CRBN led to the explosion of heterobifunctional PROTACs that recruit novel proteins of interest for proteasomal degradation [62,90,118,125]. Another class of molecular glue degrader is Indisulam, an aryl sulfonamides drug that recruits splicing protein RNF39 to DDB1-associated and CUL4-associated factor 15 (DCAF15 E3 ligase), which resulted in RNF39 ubiquitination, followed by its degradation [56,146,147,148]. As a result, Indisulam is being used as an E3 ligase recruiter for PROTAC development. Li et al. successfully demonstrated BRD4 degradation by recruiting DCAF15 in a PROTAC-dependent manner [42].

A rational design was used by Slabicki et al. to discover (*R*)-CR8 (**49, Figure 8**) as a cyclin K molecular glue degrader [149]. The glue **49** is an analog of roscovitine (**2**), which was previously reported as a CDK1/2/3/5/7/9 inhibitor [150]. A correlation of drug-sensitivity data of over 4500 drugs against 578 cancer cell lines and the expression of 499 E3 ligase components showed a positive correlation between the cytotoxicity of **49** and the mRNA levels of DDB1, a CUL4 adaptor protein. A quantitative proteome-wide mass spectrometry study showed a decrease in cyclin K in **49**-treated cells, which was rescued by the pretreatment of cells with MLN7243 (an E1 ubiquitin-activating enzyme inhibitor), MLN4924, and MG132. Biophysical, point mutation, and X-ray crystallographic studies showed that in the presence of **49**, CDK12 interacts with DDB1. Cyclin K is a binding partner of CDK12, and overlays showed that it did not make direct contact with DDB1. Interestingly, in the DDB1-CDK12-Cyclin K conformation, cyclin K was transformed into a “neo-substrate” for CUL4 and was degraded. Glue **49**-induced cyclin K degradation was blocked by co-treatment with either the inhibitor of E1 ubiquitin-activating enzyme (MLN7243) or MLN4924. CRISPR-Cas9 resistance screens demonstrated the involvement of the CUL4 complex components in mediating **49**-induced toxicity. In vitro ubiquitination assays showed that the CUL4 complex core that included RBX1-DDB1 was sufficient for **49** induced cyclin K degradation. Other CDKs (9 and 13) were also associated with DDB1 in the presence of **49**. However, these complexes did not degrade cyclin K to the same degree as the CDK12-DDB1 complex. X-ray crystallographic studies showed that the purine core of analog **49** occupied the ATP-binding site of CDK12, whereas the hydrophobic phenylpyridine ring interacted with the BPC domain of DDB1. Structure–activity relationship (SAR) studies showed that any alteration to the phenylpyridine ring resulted in diminished activity, suggesting the importance of the terminal groups that interact with DDB1 for robust cyclin K degradation. This study, along with another recent report, showed that the minor modification of a surface-exposed region of a small molecule converted it from an inhibitor to a glue [151,152], which suggests that exploiting the surface-exposed regions of CDK inhibitors could yield CDK-selective molecular glues.

## 4. Conclusions and Outlook

Novel small molecules that modulate the function or levels of specific proteins are beginning to emerge as the primary modality for cancer therapy over other strategies such as surgery, chemotherapy, and radiation therapy. In recent years, kinase inhibitors have emerged as a successful class of targeted therapeutics. Several kinase inhibitors are in clinical use or under clinical/preclinical studies. The CDK family of proteins are an important class of kinases that play prominent roles in regulating cellular transcription, mRNA processing, and the cell cycle, and thus, have become attractive therapeutic targets for cancer. The similar ATP binding sites among CDKs make the development of selective CDK inhibitors a challenging task. Even FDA-approved drugs such as palbociclib, ribociclib and abemaciclib are dual CDK4/6 inhibitors. Moreover, most of the CDKs inhibitors developed to date are non-covalent; thus, accurately estimating the dosage required to inhibit the kinase continues to vex translational scientists. PROTACs and molecular glues have emerged as an alternative approach that relies on artificially inducing degradation of the targeted protein by hijacking the cellular quality-control machinery. PROTACs and molecular glues rely on the topology of surface-exposed lysine residues for their activity and, therefore, provide a different modality to develop CDK-selective modulators. Moreover, the “catalytic nature” and “event-driven” pharmacology of PROTACs/molecular glues allows sub-stoichiometric dosing to eliminate the targeted CDK.

Here, we summarized the reported CDK selective PROTACs. Several factors such as the differential distribution of lysine residues, stable ternary complex formation, structure complementarity of CDK, and the components of the E-ligase complex dictate selective degradation. Unlike kinase inhibitors, whose efficacy is closely associated with its binding affinity, the linker length, linker composition, and the choice of the E3 ligand play a greater role in dictating the potency and selectivity of PROTACs. Nonetheless, the design of any PROTAC is an iterative process, and extensive optimization is needed to develop a potent and selective degrader. Among others, hook effects and unknown off-target effects are likely to be the growing pains as the PROTAC-based strategies mature. Most of the PROTACs developed thus far have not been evaluated for in vivo efficacy and, therefore, would require further optimization to achieve the desired pharmacokinetic (PK) profile. The successful development of prodrug **33** (Figure 7), which improved the oral bioavailability of parent PROTAC from 1 to 68%, suggests that PROTACs can be optimized for PK [81]. Presently, the PROTACs ARV-471 and ARV-110 are in Phase II clinical trials, and the PROTACs KT-474 and NX-2127 are in phase I clinical trials. These early clinical candidates provide the scientific community with the necessary drive to continue to pursue PROTACs as an alternative therapeutic strategy. Future studies will focus on developing disease-specific PROTACs either by exploring novel disease-specific POI and/or using ligands that target tissue-specific E3 ligases. Alternative strategies to achieve tissue specificity could include developing antibody PROTAC conjugates for targeted delivery to the tumors [67]. The recent discovery of molecular glues, which are small molecule inhibitors with minor modification of surface-exposed regions, for targeted protein degradation [151,152], also indicates a bright future for these therapeutic modalities that exploit the cellular protein degradation machinery.

## Figures and Tables

**Figure 1 cancers-13-05506-f001:**
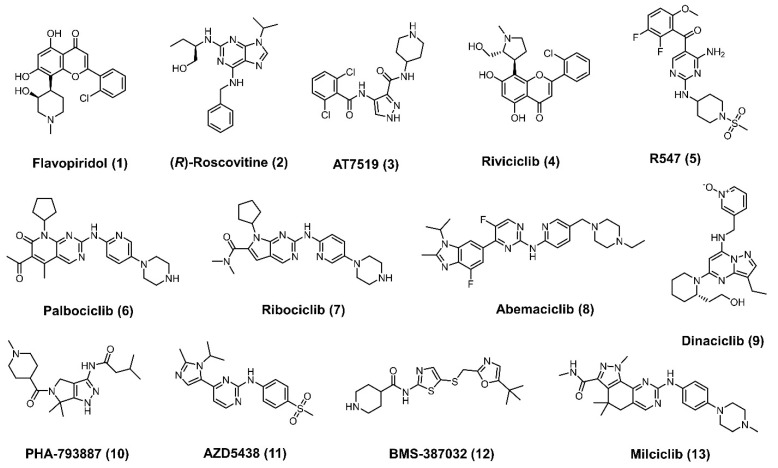
CDK inhibitors that are advanced to the clinic/clinical trials.

**Figure 2 cancers-13-05506-f002:**
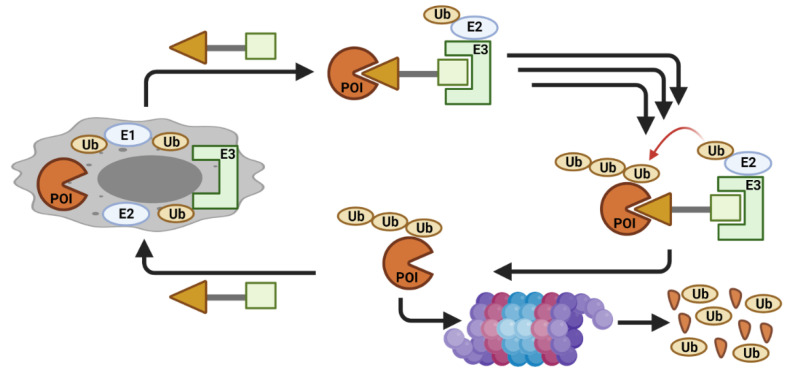
Schematic cartoon showing a PROTAC mechanism of action (created with BioRender.com).

**Figure 3 cancers-13-05506-f003:**
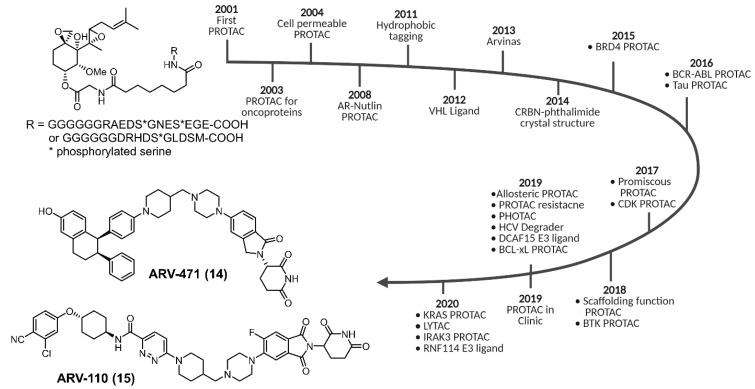
PROTACs evolution from bench to bedside (created with BioRender.com).

**Figure 4 cancers-13-05506-f004:**
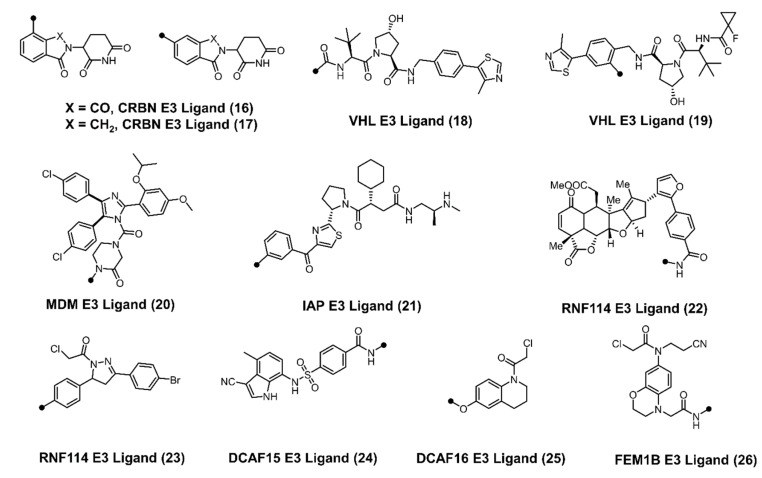
Chemical structures of different E3 ligands that have been used successfully to degrade a POI (• represent the site for linker conjugation).

**Figure 5 cancers-13-05506-f005:**
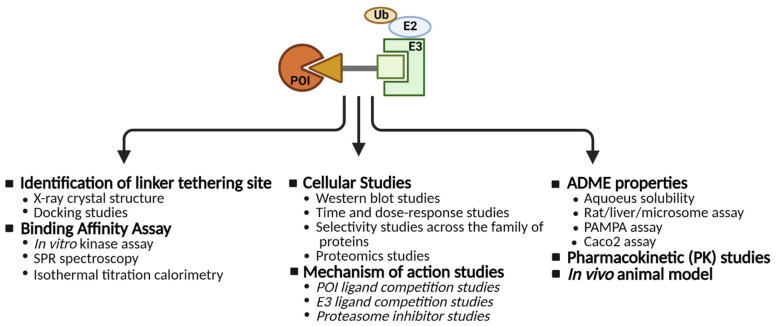
A flowchart highlighting the development of a PROTAC (created with BioRender.com).

**Figure 6 cancers-13-05506-f006:**
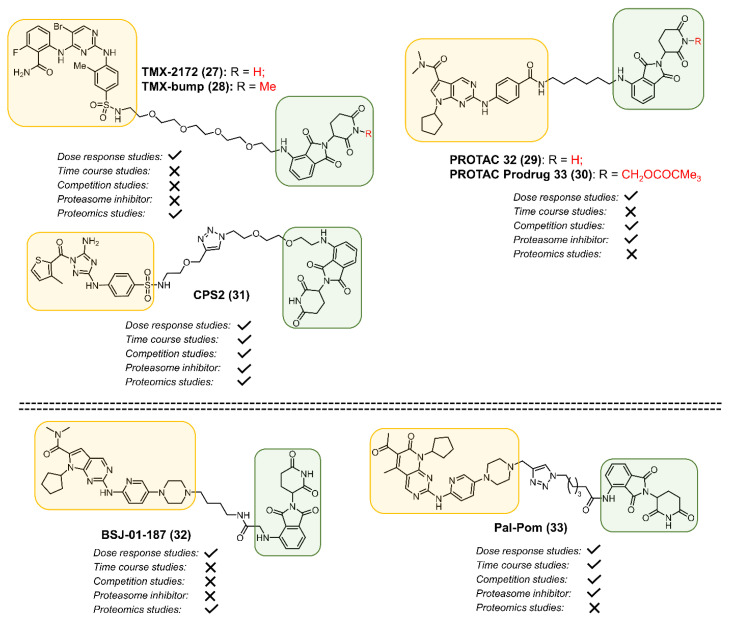
Structures of selective CDK2 (**27**, **29**, and **31**) and CDK4 (**32**, **33**) PROTACs.

**Figure 7 cancers-13-05506-f007:**
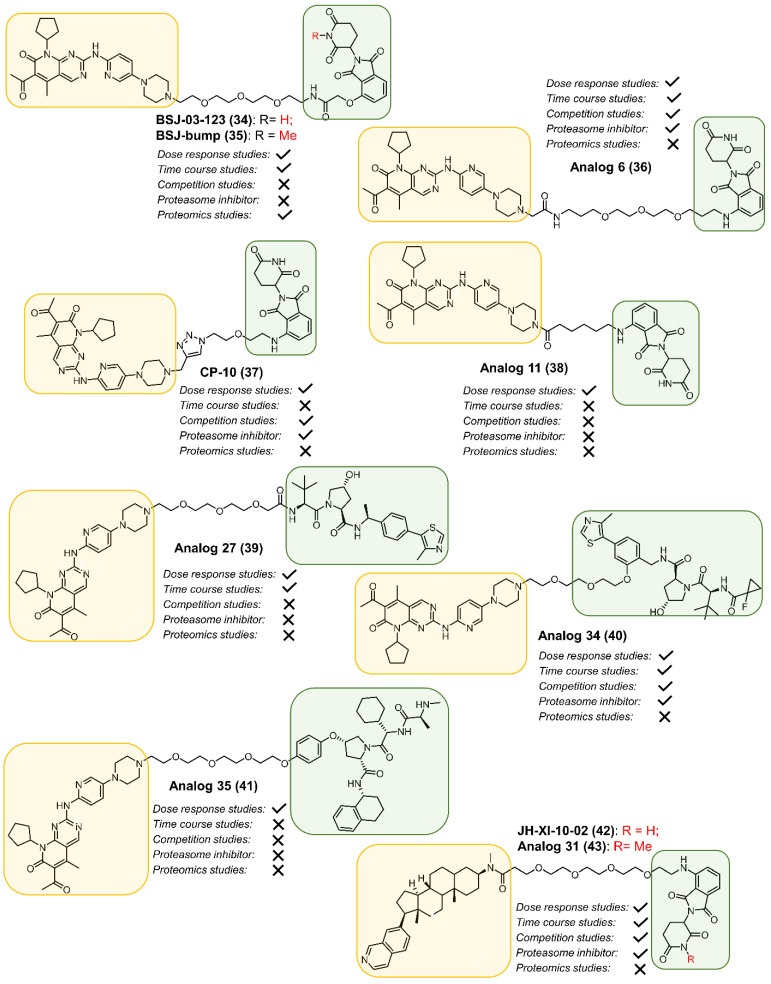
Structures of CDK6 (**34**, **36–41**) and CDK8 (**42**) PROTACs.

**Figure 8 cancers-13-05506-f008:**
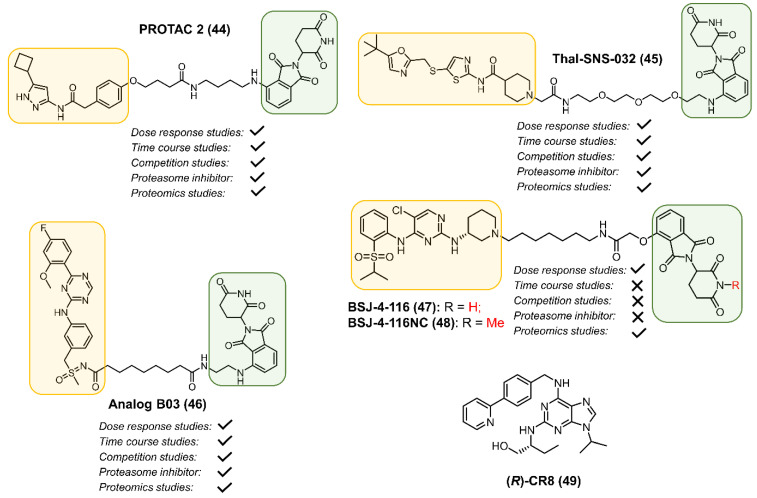
Structures of CDK9 PROTACs (**44**–**46**), CDK12 (**47**) PROTAC and a Cyclin K molecular glue (**49**).

**Figure 9 cancers-13-05506-f009:**
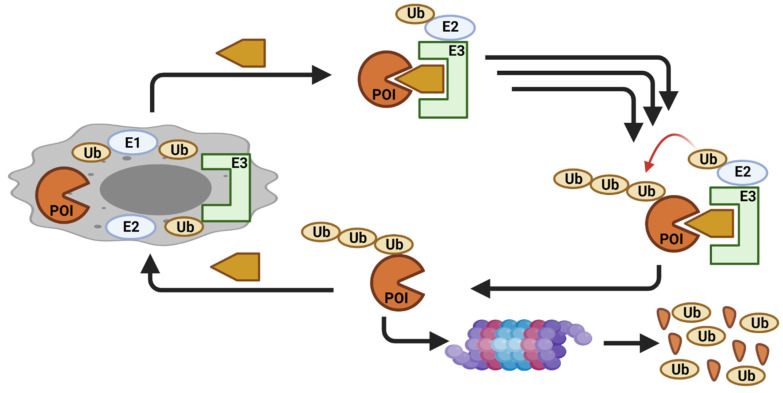
Mechanism of action of a prototypical molecular glue (created with BioRender.com).

**Table 1 cancers-13-05506-t001:** CDK inhibitors in clinical trials.

NCT No.	Conditions	Interventions	Phase
04802759 ^a^	BC	GDC-9545, Abemaciclib, Ipatasertib, GDC-0077, Ribociclib, Everolimus	I, II
04607668 ^a^	MCC	Trilaciclib	III
04585724 ^a^	BC	Abemaciclib, Palbociclib, Ribociclib	I
04553133 ^a^	SCLC, NSCLC, TNBC, EGFR2, OC	PF-07104091 monotherapy, PF-07104091 + Palbociclib, PF-07104091 + Palbociclib + Letrozole	II
04469764 ^a^	EC, OC	Abemaciclib, Anastrozole, Letrozole	II
04282031 ^a^	ST, BC	BPI-1178 ^c^, Fulvestrant, Letrozole	I, II
04247126 ^a^	ST, BC, SCLC	SY-5609 ^d^, Fulvestrant	I
04116541 ^a^	ST	HDM201, Ribociclib, Cabozantinib, Alectinib	II
04049227 ^a^	OC	Abemaciclib, Letrozole	I
04040205 ^a^	STS	Abemaciclib	II
04010357 ^a^	SCLC	Abemaciclib	II
03959891 ^a^	BC	Ipatasertib, Fulvestrant, Palbociclib	I
03740334 ^a^	ALL	Ribociclib, Dexamethasone, Everolimus	I
03675893 ^a^	EC	Letrozole, Abemaciclib	II
03593915 ^b^	MS	Alvocidib + Decitabine, or Azacitidine	I, II
03455270 ^b^	BC	G1T48, Palbociclib	I
03439735 ^a^	BC	AI and Palbociclib	II
03386929 ^b^	NSCLC	Avelumab ^e^, Axitinib, Palbociclib	I, II
03363893 ^a^	ST	CT7001, CT7001 + Fulvestrant	I, II
03310879 ^a^	Cancer	Abemaciclib	II
03285412 ^a^	BC	Ribociclib, ET	II
03280563 ^a^	BC	Atezolizumab ^f^, Bevacizumab ^g^, Entinostat, Exemestane, Fulvestrant Ipatasertib, Tamoxifen, Abemaciclib	I, II
03227328 ^a^	BC	CDK4/6 inhibitor + ET, chemotherapy + ET	II
03220178 ^a^	BC	Palbociclib, Fulvestrant, Anastrozole, Letrozole, Exemestane	IV
03170206 ^a^	LC	Binimetinib, Palbociclib	I, II
03155997 ^a^	BC	Abemaciclib, ET	III
03110744 ^a^	BC	Palbociclib	II
02712723 ^a^	BC	Letrozole, Ribociclib	II
02644460 ^a^	BT, ST, ES, BC	Abemaciclib	I
02632045 ^a^	BC	LEE-011 (Ribociclib), Fulvestrant	II
02626507 ^a^	BC	Gedatolisib, Faslodex, Palbociclib, Zoladex	I
02603679 ^a^	BC	Paclitaxel, Tamoxifen + Palbociclib, AI + Palbociclib, Goserelin + AI + Palbociclib,	II
02599714 ^b^	BC	AZD2014, Palbociclib, Fulvestrant	I
02592083 ^b^	BC	Tamoxifen, Goserelin, Palbociclib	II
02586675 ^b^	BC	Tamoxifen, Ribociclib, Goserelin	I
02503709 ^b^	ST	AT7519, Onalespib	I
02499146 ^b^	BC	Palbociclib, Letrozole	I
02095132 ^b^	BT	Adavosertib, Irinotecan Hydrochloride	I, II
01864746 ^b^	HR	Palbociclib	III
01723774 ^a^	BC	PD0332991, Anastrozole, Goserelin	II
01676753 ^b^	BC, TNBC	Dinaciclib, Pembrolizumab ^h^	I
01522989 ^b^	ST	PD-0332991, 5-FU, Oxaliplatin	I
01434316 ^a^	ST	Dinaciclib, Veliparib	I

^a^ recruiting; ^b^ active, not recruiting; ^c^ undisclosed CDK4/6 inhibitor; ^d^ undisclosed CDK7 inhibitor; ^e^ human IgG1 anti-PDL1 monoclonal antibody; ^f^ an anti-programmed death-ligand 1 antibody; ^g^ a humanized monoclonal antibody that binds to all VEGF-A isoforms; ^h^ a humanized antibody inhibiting PD-1 receptor; Non-small Cell Lung Cancer (NSCLC); Small Cell Lung Cancer (SCLC); Triple Negative Breast Cancer (TNBC); Breast Cancer (BC); Lung Cancer (LC); Myelodysplastic Syndromes (MS); Central Nervous System (CNS); Acute Lymphoblastic Leukemia (ALL); Solid Tumor (ST); Ewing Sarcoma (ES); Metastatic Colorectal Cancer (MCC); Brain Tumor (BT); Bone Cancer (BC); Endometrial Carcinoma (EC); Deficiency Disorder (DD); Soft Tissue Sarcoma (STS); Aromatase Inhibitor (AI); Endocrine therapy (ET).

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
