# Peer review of "Inhibitors, PROTACs and Molecular Glues as Diverse Therapeutic Modalities to Target Cyclin-Dependent Kinase"

_cancers, 2021, doi:10.3390/cancers13215506_

Round 1

Reviewer 1 Report

Recommendation: Publish after minor revision.

Comments:

Authors of this review discuss the research progress of CDK-based degraders and molecular glues from the different chemical structure and biological results between degraders and the related inhibitors. Overall, it is an excellent, comprehensive, timely review CDK PROTAC and molecular glues. However, there are rooms for improvements, as detailed below.

  1. The references don’t fulfill the Cancers format. Please correct the format of the references.
  2. The structural formulas of some compounds need to be corrected. For example Figure 9, 10, 13.
  3. In Figure 8, PROTAC 36 and 37 have no significant difference in the degradation of CDK4/6. Therefore, the authors cannot define them as selective CDK4 degraders.
  4. The title of section 3 is CDK degradation using molecular glues. In fact, there are no reports of molecular glues that degrade CDK. Compound 74 can induce Cyclin K degradation, but its mechanism is different from general PROTAC. Thus, I think the description in this part needs to be revised.
  5. In Figure 18, the target protein ligand of BSJ-4-116 is not THZ531, please correct the description.

Author Response

Authors of this review discuss the research progress of CDK-based degraders and molecular glues from the different chemical structure and biological results between degraders and the related inhibitors. Overall, it is an excellent, comprehensive, timely review CDK PROTAC and molecular glues. However, there are rooms for improvements, as detailed below.

Response: We would like to thank the reviewers for their time and appreciate the feedback to improve the overall quality of this manuscript.

  1. The references don’t fulfill the Cancers format. Please correct the format of the references.

Response: Per the Cancers requirements, any ACS reference format is acceptable.

  1. The structural formulas of some compounds need to be corrected. For example Figure 9, 10, 13.

Response: We regret the errors and thank the reviewer for identifying them. We have corrected them in the revised manuscript.

  1. In Figure 8, PROTAC 36 and 37 have no significant difference in the degradation of CDK4/6. Therefore, the authors cannot define them as selective CDK4 degraders.

Response: We thank the reviewer for bringing this to our attention. The pal-pom PROTAC is ~3-fold selectivity for CDK4 over CDK6 (please see DC50 values in Table 1). In the revised manuscript we have removed the structure of rib-pom and edited the text to specifically state the above.

  1. The title of section 3 is CDK degradation using molecular glues. In fact, there are no reports of molecular glues that degrade CDK. Compound 74 can induce Cyclin K degradation, but its mechanism is different from general PROTAC. Thus, I think the description in this part needs to be revised.

Response: This was a careless oversight on our part and we agree with the reviewer that there are no known molecular glues that degrade CDK. In the revised manuscript the section title reads “Perturbing CDK function by a molecular glue”

  1. In Figure 18, the target protein ligand of BSJ-4-116 is not THZ531, please correct the description.

Response: We regret the error and have revised the section accordingly.

Reviewer 2 Report

The topic of the Review Article “Inhibitors, PROTACs and Molecular Glues as Diverse Therapeutic modalities to Target Cyclin Dependent Kinase”  is interesting but in order to provide a good overview of the subject I suggest a complete revision and reorganization of the text.

The authors should focus on novel therapeutic approaches while some unnecessary mechanisms and compounds are listed. Too many figures and few legend descriptions make the review  dispersive. I would encourage the authors to improve the manuscript making it shorter and more usable with “take-home messages” described more clear.

Author Response

The topic of the Review Article “Inhibitors, PROTACs and Molecular Glues as Diverse Therapeutic modalities to Target Cyclin Dependent Kinase”  is interesting but in order to provide a good overview of the subject I suggest a complete revision and reorganization of the text. The authors should focus on novel therapeutic approaches while some unnecessary mechanisms and compounds are listed. Too many figures and few legend descriptions make the review  dispersive. I would encourage the authors to improve the manuscript making it shorter and more usable with “take-home messages” described more clear.

Response: We appreciate the constructive critique of the manuscript. Per the reviewers recommendation, we have extensively revised and reorganized the manuscript. In the revised manuscript we have condensed the figures to highlight only the key selective PROTACs and included the characterization data for the PROTACs in the figures. We have also eliminated redundancies with the characterization of the PROTACs and included a single paragraph early in the manuscript describing the characterization. We have compared the structural commonalities and differences within each set of PROTACs and have summarized the progress in developing selective PROTACs for each CDK. We hope these changes are in line with the reviewers vision for improving the quality of the manuscript.

Reviewer 3 Report

This is a very comprehensive review of PROTACs and Molecular Glues from authors who have track

record of published work in this field . Minor comment is about the descriptions of individual CDK PROTACs .

This can be condensed to a able or highlight the similarities, differences or salient points and perhaps figures describing individual structures can be added as supplemental file. Readers may appreciate broad overview and condensed , succinct points to understand the progress in the field and refer to specific publications to learn more details about each PROTAC .

Overall, this is very well written review that will interest readers of this journal.

Author Response

This is a very comprehensive review of PROTACs and Molecular Glues from authors who have track record of published work in this field . Minor comment is about the descriptions of individual CDK PROTACs. This can be condensed to a able or highlight the similarities, differences or salient points and perhaps figures describing individual structures can be added as supplemental file. Readers may appreciate broad overview and condensed , succinct points to understand the progress in the field and refer to specific publications to learn more details about each PROTAC . Overall, this is very well written review that will interest readers of this journal.

Response: We appreciate the constructive critique of the manuscript. Per the reviewers recommendation, we have extensively revised and reorganized the manuscript. In the revised manuscript we have condensed the figures to highlight only the key selective PROTACs and included the characterization data for the PROTACs in the figures. We have also eliminated redundancies with the characterization of the PROTACs and included a single paragraph early in the manuscript describing the characterization. We have compared the structural commonalities and differences within each set of PROTACs and have summarized the progress in developing selective PROTACs for each CDK. We hope these changes are in line with the reviewer's vision for improving the quality of the manuscript.

Round 2

Reviewer 2 Report

The authors modified according almost all the recommendations of the reviewers.